# Destructive Interference of the Superconducting Subband Condensates in the Quasi-1D Multigap Material Nanostructures

**Wojciech Julian Pasek *** **, Marcos Henrique Degani and Marcelo Zoéga Maialle**

Faculdade de Ciências Aplicadas, Universidade Estadual de Campinas, R. Pedro Zaccaria 1300,
Limeira 13484-350, SP, Brazil
***** Correspondence: wjpasek@unicamp.br or wojciech.julian.pasek@gmail.com

**Abstract:** This modelling work concerns the effects of the interference between two partial subband condensates in a quasi-one-dimensional superconducting superlattice. The iterative under-relaxation with phase control method is used to solve Bogoliubov–de Gennes equations in the envelope ansatz. This method—easily generalisable to a wide class of multiband superconducting systems—allows us to obtain both the constructive and the destructive interference solution. The discussion is centred on the latter case, with one of the condensates collapsing with increased inter-subband coupling strength, due to the other—the dominating one—imposing its symmetry on the overall order parameter. The in-depth qualitative analysis is made of underlying intra-subband and inter-subband dynamics, such as the possible factors determining the dominant subband condensate or the ones determining the region where the destructive solution coexists with the constructive one. A comprehensive discussion with the recent works concerning inter-band coupling effects follows, pointing that the destructive solution is nearly universally omitted.

**Keywords:** Bogolubov–de Gennes equations; multiband superconductivity; condensate interaction



## 1. Introduction

There is a long and well-established history of research employing the effective mass method (envelope ansatz) in the study of quantum size effects in superconducting nanostructures. The naturally occurring superlattice (SL) of stripes was suggested as a model for the cuprate perovskites already in [1], with continuation in [2] reporting a Feshbach–Fano (FF) resonance. The quantum size effects and related resonant processes in nanowires of a $\mu$m length and nm diameter were studied in [3–6]. The superconducting nanofilms were theoretically investigated as early as 1963, when the well-known work [7]. Analogous methods were employed in later works, including [8,9]. Similar systems were also studied in the recent years, e.g., [10–12]. The effects of quantum confinement in the step-like nanosystems were under investigation in [13,14]. Other recent works consider advanced methods of the topological resonance-driving of SL systems, such as the Rashba spin coupling in [15,16] and applying pressure to the doped SL systems in [17,18]. The two latter works develop the earlier ideas presented in, e.g., [2,19,20].

Nanosystems of reduced dimensionality with strong coupling are vulnerable to order parameter fluctuations effectively suppressing the superconductivity. However, as discussed in the works [21,22], with the help of a Ginzburg–Landau-like model, said fluctuations can be screened in multiband systems by a relatively small coupling of a shallow band with strong pairing and a deep band with weak pairing, recovering the mean field result. The very recent work [23] addressed the topic of complex superconducting condensates, to the formation of which contribute both quasi-one-dimensional (Q1D) bands as well as the "reservoir condensates" of higher dimensionality energy bands.

Our previous works [24,25] studied the Q1D SLs to be realised in materials with superconductivity of multiband character. The present work is focused on these solutions

of the Bogoliubov–de Gennes equations (BdG) in which the interaction between the condensates corresponding to two different material subbands is destructive in character. As a general rule, in the case of a single condensate, the global phase $\phi$ of the order parameter $\Delta \exp i\phi$ is arbitrary in BdG. The same is true for two uncoupled condensates. In the case of a system containing two condensates A and B coupled in a way that can be called, e.g., *exchange-like inter-band pairing* [26] or *Josephson-like pair transfer* [27], the system is invariant under the global phase shift applied to both order parameters, $\Delta_A \rightarrow \Delta_A \exp i\phi_1$ and $\Delta_B \rightarrow \Delta_B \exp i\phi_1$, but it depends on relative phase between the A and B. In the absence of the magnetic field, the order parameters can be defined as real quantities. This leaves two possible relative phases of the band condensates in the solutions of BdG: of the same or of the opposite signs. Correspondingly, the interaction of the condensates can be either constructive or destructive.

In the previous works describing the effects of the multiband interactions, most focus is given to the constructive interactions, while the destructive ones are typically omitted. A notable exception is the work [26], which is discussed at length in Appendix C.

In [25], the single-particle energies of the envelope ansatz model were fitted to the few-monolayer $MgB_2$ dispersions as described by the experiments and ab initio calculations. In this kind of material, due to the few-monolayer thickness, the usual $\sigma$ and $\pi$ bands of the bulk $MgB_2$ split to form a series of subbands each ($\sigma_{L1}, \sigma_{L2}, \sigma_{U1}, \sigma_{U2}, \ldots, \pi_1, \pi_2, \ldots$). The quantum size effects of the SL itself lead to further splitting and the formation of a set of Q1D minibands for each of the mentioned subbands. While the mentioned work focused on the formation of single-subband condensates in such a system, here the model is purposefully simplified in order to sharply focus on the basic dynamics of the destructive interference in an easy-to-interpret theoretical environment. The application of the same method to a more sophisticated model would be the topic of forthcoming work.

## 2. Materials and Methods

The system under investigation is a periodic SL, transversally confined and consisting of alternating constricted and unconstricted parts in the longitudinal direction. The constriction is introduced by the imposition of the prohibitively large potential energy in the constriction barrier, as seen in Figure 1a. Originally [25], the kinetic parts of the single-particle energies were modelled on the few-monolayer $MgB_2$, which is also true in the present work. Here, only two subbands were taken into account, called $\sigma_{L1}$ and $\sigma_{L2}$, and only one miniband for each of the subbands was included, the one closest to the Fermi energy. The minibands represent the single-particle energies at each point of the SL Brillouin zone (SLBZ), with the SL longitudinal wavenumber $Q$ (not to be confused with the *material* Brillouin zone of the underlying crystal structure). However, while in the mentioned previous work the subband condensates were non-interacting, here we introduce the coupling between them as governed by the inter-subband coupling constant $J_1$, which has the dimension of energy and is a natural counterpart to the intra-subband coupling constant $J_0$. To summarise, the coupling between the different parts of the SLBZ for the same miniband are included, the miniband–miniband coupling in the scope of the same subband is omitted for simplicity, while the $\sigma_{L1} \leftrightarrow \sigma_{L2}$ coupling is taken into account. As was mentioned in Section 1, the goal of this definition of the system is to expose the underlying mechanism of the condensate interaction in an environment where relatively simple but in-depth interpretations can be made.

For the details of the geometrical definition and parametrisation of the model at the single particle level, the reader is directed to Sections 3.1–3.3 of [25]. The single-particle miniband spectrum is shown in Figure 1c for $\sigma_{L1}$ and in Figure 1d for $\sigma_{L2}$. In this work, only miniband 2 of Figure 1c—thick red line—and miniband 5 of Figure 1d—thick green line—are taken into account, as explained above. Furthermore, the first case to be studied has an additional simplification, with the Fermi energy shifted by $+11.6$ meV— corresponding to the horizontal dash-dotted line and marked as $E_F'$—in order to lie in the middle of the miniband 5 of $\sigma_{L2}$. This way, the system is composed in the single-particle

picture of two different metallic subbands (henceforth called the *double-metallic system* or M-M). After that, the default system without this shift is studied, which is a mixed metallic/semiconductor or insulating-like system (henceforth called the *metallic-insulating system* or M-I). This way, it is possible to illustrate how the additional insulating-like gap qualitatively changes the characteristics of the system.

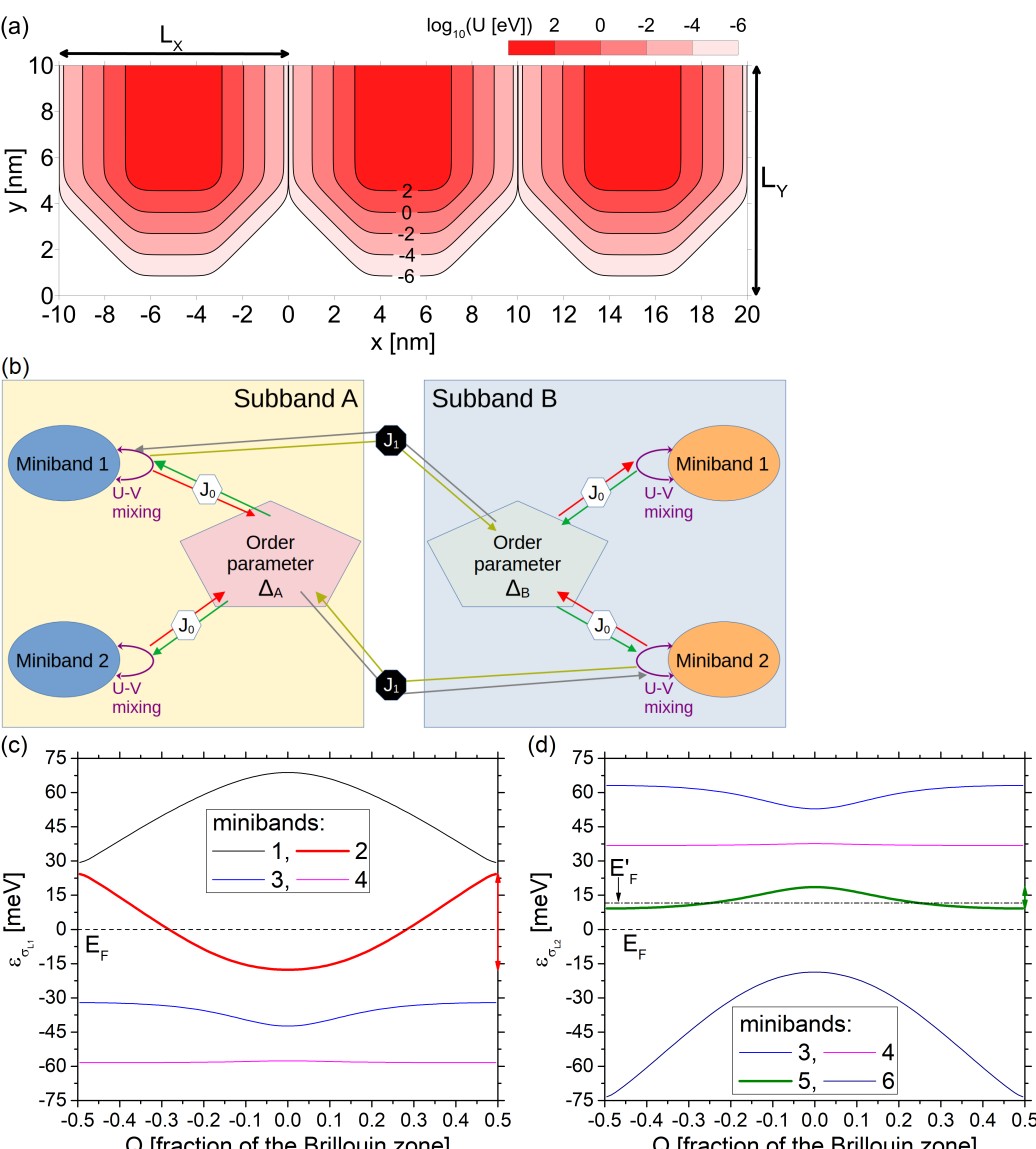

**Figure 1.** (**a**) SL geometry, showing three consecutive primary cells and the $\log_{10}$ of the potential energy in eV. (**b**) Schematic general illustration of the model. Note that in this work only one miniband per each subband is taken into account. (**c**) The single-particle miniband energy $\epsilon$ spectrum for the $\sigma_{L1}$ subband. The thick line shows the miniband included in the model. (**d**) The same as (**c**) but for the $\sigma_{L2}$ subband. The shifted $E_F$ for the M-M (see text) is shown by the horizontal dash-dotted line.

The BdG model differs from the previous work, as described above. It is developed in the scope of envelope ansatz and the Anderson approximation, with the self-consistent under-relaxation iteration of the order parameter matrix elements. The phase control of the condensates is introduced both for the initial random guess and at each iterative step. The details can be found in Appendix A. Most critically, there are two order parameters $\tilde{\Delta}_{\sigma_{L1}}$ and $\tilde{\Delta}_{\sigma_{L2}}$ describing the $\sigma_{L1}$ and $\sigma_{L2}$ condensates. Here, the $\sim$ signifies that they are (dimensionless) quantities corresponding to partial one-subband condensates, not the overall order parameter as experienced by any of the two subbands. From the point of

view of the $\sigma_{L1}$ subband, the overall order parameter—the quantity corresponding to the off-diagonal term in BdG—is $\Delta_{\sigma_{L1}} = J_0 \tilde{\Delta}_{\sigma_{L1}} + J_1 \tilde{\Delta}_{\sigma_{L2}}$, whereas from the point of view of the $\sigma_{L2}$ subband, it is $\Delta_{\sigma_{L2}} = J_0 \tilde{\Delta}_{\sigma_{L2}} + J_1 \tilde{\Delta}_{\sigma_{L1}}$. In specific, this means that the inter-subband phonon coupling strengths of the two subbands are relatively similar and can both be represented by a single parameter $J_0$ and that the inter-subband $\sigma_{L1} \leftrightarrow \sigma_{L2}$ coupling is symmetric and can be represented by a single parameter $J_1$. The general scheme of the model is shown in Figure 1b.

## 3. Results

### 3.1. The Double-Metallic System

The results for the M-M system are shown in Figure 2. The dependence of the energy gaps on the inter-subband coupling strength $J_1$ for the two subbands is shown in (a)—$\sigma_{L1}$ gap or $G_{\sigma_{L1}}$—and in (b)—$\sigma_{L2}$ gap or $G_{\sigma_{L2}}$—for three chosen values of the intraband parameter $J_0$ (the black line for 10 meV, the red line for 11 meV and the blue line for 12 meV, respectively). The three lines follow the same general trend, with the initial value at $J_1 = 0$—uncoupled condensates—increasing with the relative inter-subband coupling strength $J_0$, as expected. The dashed lines correspond to the case where the condensates have the same phase and hence interact constructively. In this case, a simple linear increase in the gap as $J_1$ increases is observed in both (a) and (b). The solid lines, on the other hand, correspond to the situation where the phase of the two condensates is opposite and thus they interact destructively. Starting from $J_1 = 0$, a linear decrease in the both gaps is encountered at first. Eventually, however, in all three cases the $G_{\sigma_{L1}}$ in (a) gap drops rapidly to zero, while the $G_{\sigma_{L2}}$ gap in (b) reaches a minimum a moment before this happens, and then increases towards the value it had for the uncoupled case. Note that the results are missing the part in the immediate vicinity of the point where the $\sigma_{L1}$ condensate collapses. This is due to the fact that the behaviour of the $G_{\sigma_{L1}}$ as it approaches zero is of the $\sim (\pm x - x_0)^c$ kind, with $c < 1$, typical for many quantities near the vicinity of the critical point of a phase transition. At the critical point, the (negative) derivative approaches infinity, which in practical terms means that, in order to obtain approximately equal drop interval of $G_{\sigma_{L1}}$, progressively exponentially denser mesh of $J_1$ values must be employed. Furthermore, in each of the self-consistent iterations, a progressively smaller value of the under-relaxation parameter $\alpha$—as defined in Appendix A—needs to be used, leading to the even faster growth of the required computational time. Consequently, at the limit of practical computational complexity, the $a_1(-J_1 - a_2)^{a_3}$ functions were instead fitted to the several last computed values. These are shown in (a) as the dotted curves.

Figure 2c,d show the $Q$-dispersion of the BdG solutions for the three chosen $J_1$ values, for $\sigma_{L1}$ and $\sigma_{L2}$, respectively. These values correspond to the A, B and C points on the $J_1 = 11$ meV lines in (a) and (b). The BdG solutions are shown as the symbols. The electron-like and hole-like single-particle energies are marked with the solid red line and the dashed blue line, respectively. The $G_{\sigma_{L1}}$ is located at the points in the SLBZ named $q_1$, where the single-particle miniband energy crosses the Fermi level. This is usually—but not always—the case, as described in [24]. The gap itself is represented with the black vertical arrow in the C case. The BdG spectrum is shifted the most strongly from the single-particle energy at $q_1$ and precisely this shift is what determines the value of the $G_{\sigma_{L1}}$ in (a). The A case corresponds to the uncoupled subbands scenario, with the greatest superconducting shift over the whole SLBZ. In the B case, the $G_{\sigma_{L1}}$ is already out of the initial linear regime, but with still a significant nonzero value. Finally, in the C case, the $G_{\sigma_{L1}}$ is on the verge of abrupt decrease to zero, with the simultaneous collapse of the corresponding condensate. The A→B→C dynamic in (c) pictures the monotonic collapse of the whole BdG miniband to the electron/hole-like spectrum. In the case of the gap for the $\sigma_{L2}$ subband $G_{\sigma_{L2}}$, shown in (d), the dynamic is different. Here, the B case also corresponds to the point out of the linear regime, but where the $G_{\sigma_{L2}}$ value is minimal, while for B→C the value of the gap increases, as shown in (b). Hence, in (d), the BdG miniband for the B case lies wholly, if slightly, below the one for the C case. The $G_{\sigma_{L2}}$ is located at the $q_2$ points in SLBZ, which are not

aligned with the $q_3$ points where the single-particle miniband crosses the $E_F$, in contrast to the $G_{\sigma_{L1}}$ case.

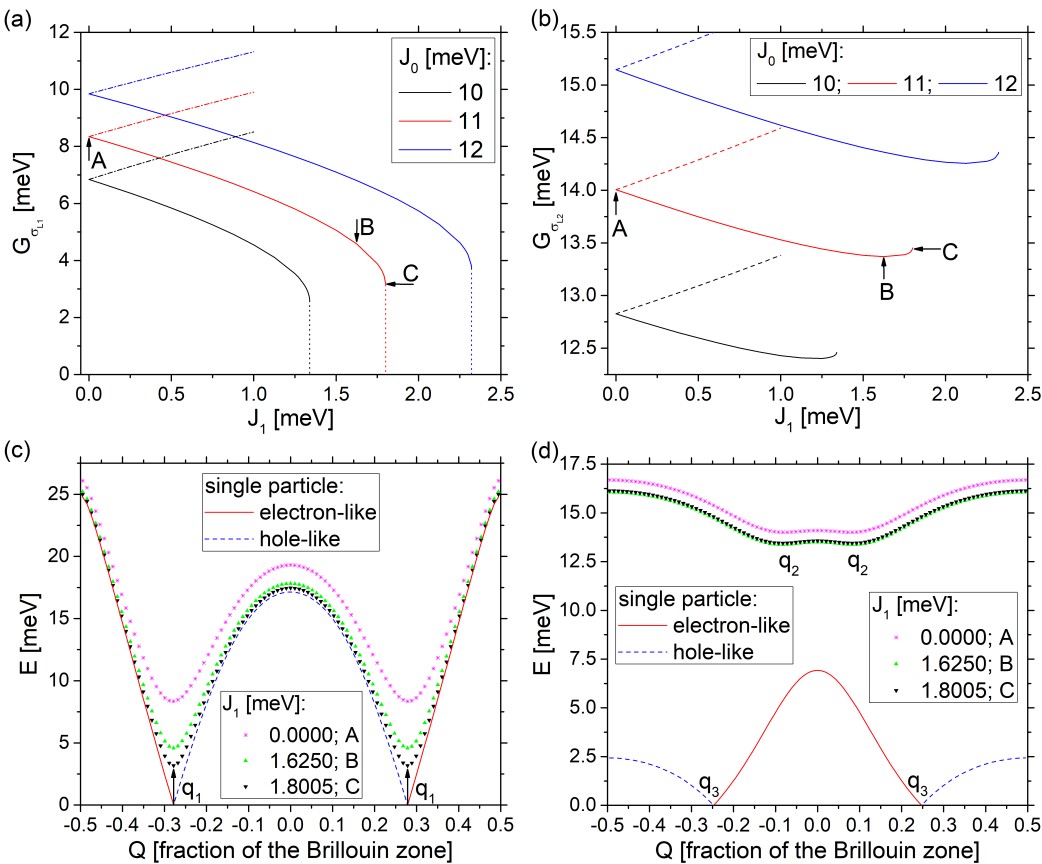

**Figure 2.** Results for the M-M system . (**a**) The $\sigma_{L1}$ gap for chosen values of $J_0$, as a function of $J_1$. (**b**) The same as (**a**) but for the $\sigma_{L2}$ gap. (**c**) The $Q$-dispersion for the $\sigma_{L1}$ miniband for $J_0 = 11$ meV and chosen $J_1$ values, corresponding to points A, B and C in (**a**,**b**). (**d**) The same as (**c**) but for the $\sigma_{L2}$ gap.

The linear regime in (b) and (c) can be intuitively interpreted, both in the case of the constructive interaction increase and in the case of the destructive interaction decrease, as a situation where the individual partial subband condensates $\tilde{\Delta}_{\sigma_{L1}}$ and $\tilde{\Delta}_{\sigma_{L2}}$ remain quasi-constant. Then, all the dynamic of the off-diagonal term is caused by the change of the $J_0$ and $J_1$ parameters. Recall that the diagonal terms of BdG are the electron-like and hole-like parts, equal to the plus/minus single-particle energy $\pm\epsilon$. Therefore, the diagonal part is zero at the crossing point $q_1$. The $\sigma_{L1}$ energy gap is then $G_{\sigma_{L1}} = |J_0\tilde{\Delta}_{q_1,\sigma_{L1}} + J_1\tilde{\Delta}_{q_1,\sigma_{L1},\sigma_{L2}}|$— see Equations (A6)–(A8), which is a linear function of $J_1$ if $J_0$ is kept constant (and vice versa) and the partial condensate elements remain quasi-constant. If the phase of $\tilde{\Delta}_{\sigma_{L1}}$ is assumed to be positive, then the sign of the derivative is the phase of the $\tilde{\Delta}_{\sigma_{L2}}$, which corresponds to the constructive or destructive interference of the condensates. In the case of $G_{\sigma_{L2}}$, the dependence is not as strict, due to the misalignment of $q_2$ and $q_3$ (cf. Figure 2d), however the general idea is the same.

In order to explain the behaviour of the non-linear part, one should investigate the symmetry of the order parameter itself. The momentum space maps $\hat{\Delta}_{n_X,n_Y,S}$ for M-M, described in detail in Appendix B are shown in Figure 3. The LHS column corresponds to the $\sigma_{L1}$ condensate, while the RHS one to the $\sigma_{L2}$ condensate. The rows correspond top-to-bottom to the points A, B and C in Figure 2a–d, respectively. The maps are spanned over the space of the allowed momentum basis functions, as dictated by the periodic/hardwall boundary conditions for $x$ or $y$, and numerated by $n_{k_x}$ and $n_{k_y}$, respectively. The pictures show only the small relevant subspace, where all of the dominating components are.

The colour scale of the pictures is as follows: the largest absolute value corresponds to pure red, its opposite to pure blue, and zero is white. In the A case, the subbands are uncoupled, so the order parameter in the top row is in fact the pure $\tilde{\Delta}_{\sigma_{L1}}$ in (a) and pure $\tilde{\Delta}_{\sigma_{L2}}$ in (b). Indeed, the phase of the order parameter in (a) is purely positive, and in (b) purely negative. This remains the case in the whole linear regime. Recall that the B case is the one where both the $G_{\sigma_{L1}}$ and $G_{\sigma_{L2}}$ are already outside of the linear regime, with the latter one at the minimum. As shown in (c), the order parameter symmetry for the $\sigma_{L1}$ subband is already broken, with the $\tilde{\Delta}_{\sigma_{L2}}$ phase and symmetry partially imposed over the $\tilde{\Delta}_{\sigma_{L1}}$ ones. On the other hand, there is almost no change in the order parameter for the $\sigma_{L2}$ subband, as shown in (d). Finally, the results in the bottom row (the C case) are almost identical to the ones in the middle row, with the phase/symmetry erosion of the (e) case somewhat more advanced. To sum up, the phase and momentum space symmetry of the order parameter of the $\sigma_{Lq}$ subband is qualitatively redefined, which leads at first to the deviation from the linear regime of the $G(J_1)$ dependence, and subsequently to the closing of the $\sigma_{L1}$ gap and collapse of the corresponding condensate. Conversely, the phase and the momentum symmetry of the order parameter for the $\sigma_{L2}$ subband is nearly unaffected by $J_1$, and therefore by the corresponding coupling between the condensates, so the non-monotonic behaviour of the $G_{\sigma_{L2}}$ can be understood as a secondary quantitative effect. To be more specific, when the $\sigma_{L1}$ condensate very rapidly proceeds to collapse, the decrease in the absolute value of the second factor in the relevant minor destructive component, $J_1\tilde{\Delta}_{q_2,\sigma_{L2},\sigma_{L1}}$, happens much faster than the increase of $J_1$.

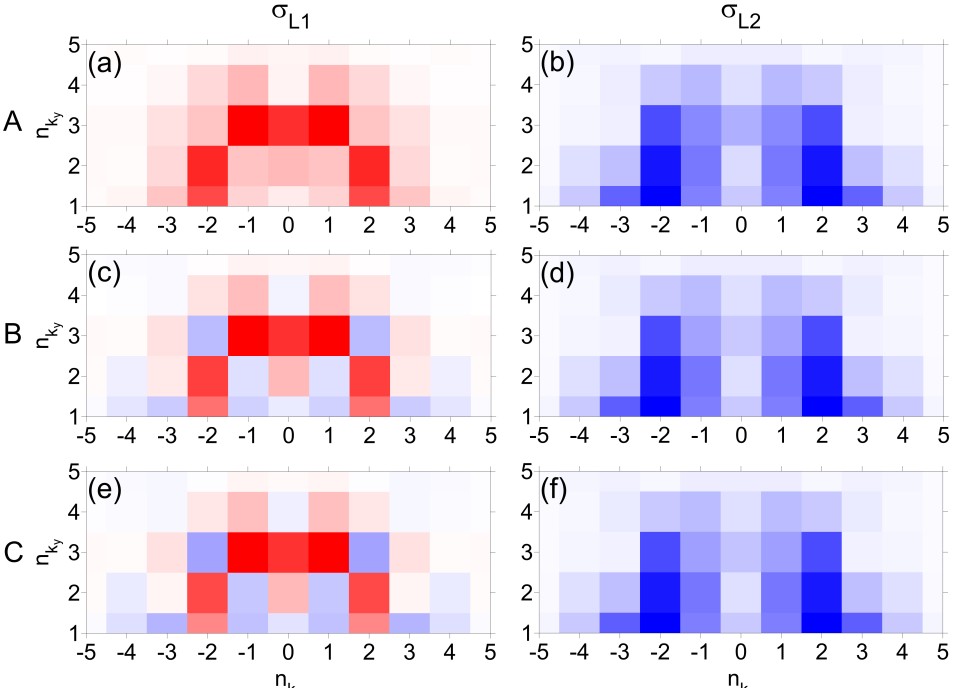

**Figure 3.** The momentum space $\Delta$ maps $\hat{\Delta}_{n_X,n_Y,S}$, for the M-M and $J_0 = 11$ meV. (**a**) The $\hat{\Delta}_{n_X,n_Y,\sigma_{L1}}$ map, where $J_1$ corresponds to the A case of Figure 2. (**b**) $\hat{\Delta}_{n_X,n_Y,\sigma_{L2}}$, A case. (**c**) $\hat{\Delta}_{n_X,n_Y,\sigma_{L1}}$, B case. (**d**) $\hat{\Delta}_{n_X,n_Y,\sigma_{L2}}$, B case. (**e**) $\hat{\Delta}_{n_X,n_Y,\sigma_{L1}}$, C case. (**f**) $\hat{\Delta}_{n_X,n_Y,\sigma_{L2}}$, C case. Each map is colour-coded so that the largest absolute value corresponds to pure red, its opposite to pure blue, and zero is white.

The reason why the competition to dominate the characteristics of the system is won by the $\sigma_{L2}$ condensate over the other one can be easily identified with the relation of the bandwidths of the corresponding single-particle minibands. The bandwidths are marked in Figure 1c,d on the RHS scale by the vertical arrows of the same colour as the corresponding minibands. They are equal to 42 meV in case of the $\sigma_{L1}$ and to 9.4 meV in the case of $\sigma_{L2}$, making the latter one approximately 4.5 times smaller. The bandwidths are inversely corre-

lated with the DOS, as the equal number of states is dispersed over a larger energy interval, which leads to the relatively stronger ability of the $\sigma_{L2}$ to generate superconductivity.

### 3.2. The Metallic-Insulating System

This section describes the study of the original system without the $E_F \to E_F'$ shift for $\sigma_{L2}$, see Figure 1d. In this case, none of the minibands of the $\sigma_{L2}$ subband crosses the Fermi level. The miniband 5, which is the closest one to $E_F$ is included in the model. Consequently, in the single-particle system, there already exists an insulating-like or semiconductor-like gap of $G_{\sigma_{L2}}^0 = 9.2$ meV at $Q = q_5$ on the edges of the Brillouin zone. The total gap in the case of this subband is $G_{\sigma_{L2}} = G_{\sigma_{L2}}^s + G_{\sigma_{L2}}^0$, with the $G_{\sigma_{L2}}^s$ being the additional gap introduced by the superconductivity, as illustrated with the two vertical arrows in Figure 4d. The $G_{\sigma_{L2}}^0$ itself is trivial—it does not depend on any superconducting parameter—and thus the behaviour of $G_{\sigma_{L2}}^s$ will be investigated.

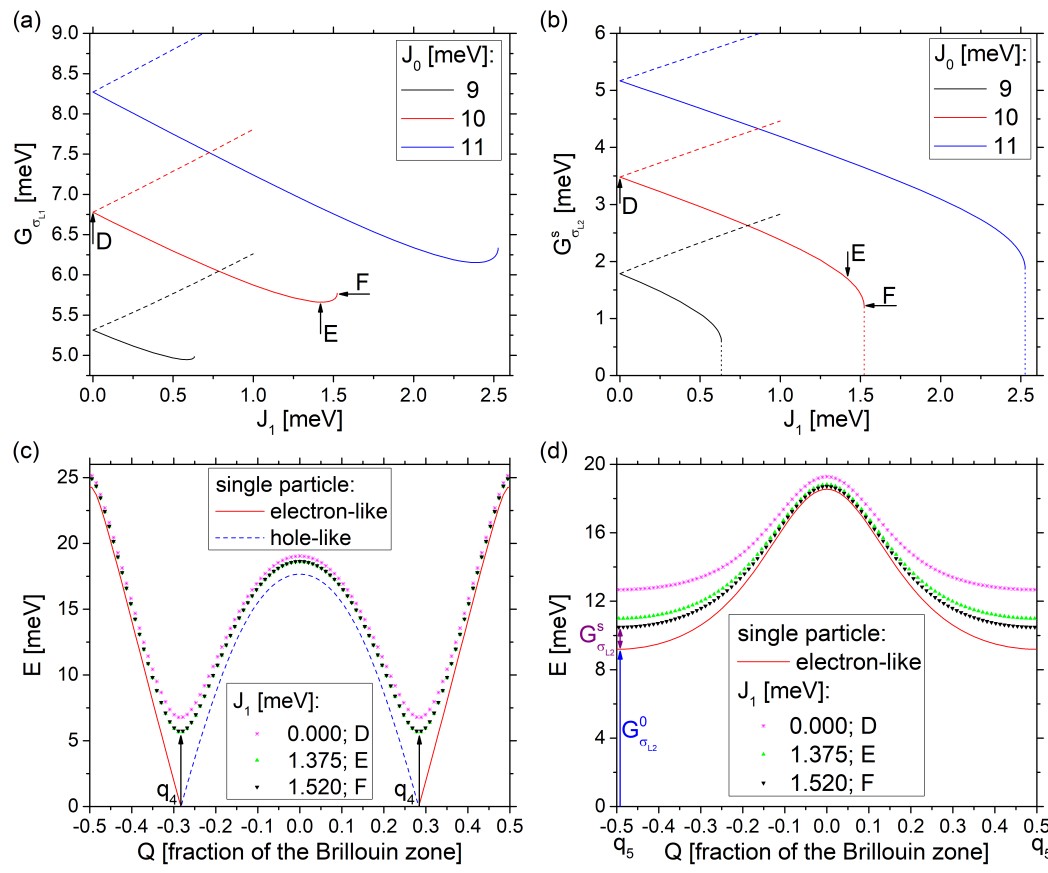

**Figure 4.** The results for the M-I system . (**a**) The $\sigma_{L1}$ gap for chosen values of $J_0$, as a function of $J_1$. (**b**) The same as (**a**), but for the $\sigma_{L2}$ gap. (**c**) The $Q$-dispersion for the $\sigma_{L1}$ miniband for $J_0 = 10$ meV and chosen $J_1$ values, corresponding to points D, E and F in (**a**,**b**). (**d**) The same as (**c**), but for the $\sigma_{L2}$ gap.

Figure 4 is analogous to Figure 2, with the M-I taken into consideration this time. The $J_1$ dependence in this system is similar to the one observed previously, but the dynamics of the subbands are swapped. In addition, the linear regime is present for sufficiently small $J_1$ and then one of the gaps rapidly drops to zero. In this instance, it is the $G_{\sigma_{L2}}^s$ one, as can be observed in (b). The other one—$G_{\sigma_{L1}}$ in (a)—first reaches a minimum and then increases towards the value it has for $J_1 = 0$. For comparison, three curves are shown in (a) and (b), corresponding to the $J_0 = 9, 10$ and $11$ meV cases, respectively. The $Q$-dispersion of the $J_0 = 10$ meV system in the points D, E and F is shown in (c) in the case of $\sigma_{L1}$ subband and in (d) for the $\sigma_{L2}$ subband. Again, the dynamic of the subbands is swapped here, with the $\sigma_{L2}$ dispersion collapsing to the single-particle dispersion. In this instance,

it tends to the pure $\epsilon$ electron-like dispersion, as the $-\epsilon$ hole-like part is absent from the Debye window ($0 < E < E_D = 75$ meV) due to the lack of crossing with the $E_F$. The BdG dispersion reaches values of $\epsilon$ much quicker in comparison to the comparatively large shift remaining at $q_5$ even for the F case, just before the sudden drop to zero, in analogy to what happened in Figure 2c. On the other hand, $G_{\sigma_{L1}}$ in Figure 4c remains open at $Q = q_4$ for all $J_1$, reaching the minimum in the E case. The D→E→F dynamic of the order parameter, shown in Figure 5 is also analogous to the A→B→C of Figure 3, keeping in mind the swap of the subband dynamics. In the M-I case, the phase and momentum space symmetry of the $\sigma_{L1}$ is imposed on the other subband outside the linear regime, as seen in (d) and (f).

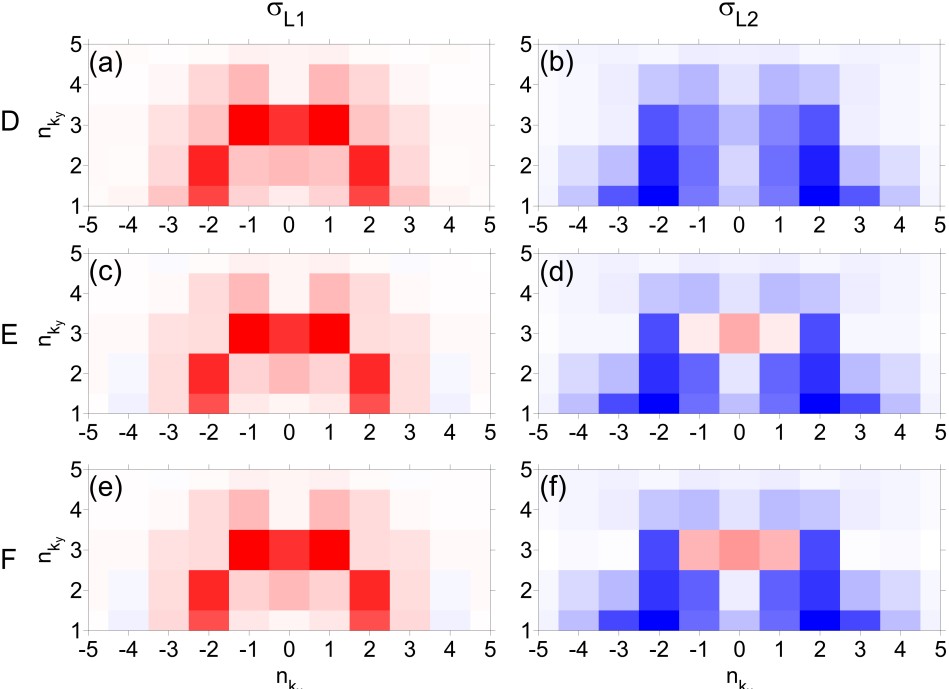

**Figure 5.** The momentum space $\Delta$ maps for the M-I, where $J_0 = 10$ meV. (**a**) The $\hat{\Delta}_{n_X,n_Y,\sigma_{L1}}$ map, where $J_1$ corresponds to the D case of Figure 2. (**b**) $\hat{\Delta}_{n_X,n_Y,\sigma_{L2}}$, D case. (**c**) $\hat{\Delta}_{n_X,n_Y,\sigma_{L1}}$, E case. (**d**) $\hat{\Delta}_{n_X,n_Y,\sigma_{L2}}$, E case. (**e**) $\hat{\Delta}_{n_X,n_Y,\sigma_{L1}}$, F case. (**f**) $\hat{\Delta}_{n_X,n_Y,\sigma_{L2}}$, F case. Each map is colour-coded so that the largest absolute value corresponds to pure red, its opposite to pure blue, and zero is white.

The change of the dominating condensate between the M-M and the M-I cases can be understood as a consequence of the distancing of the DOS involved in the creation of the $\sigma_{L2}$ condensate from the $E_F$, which takes place in the latter case. As a consequence, in a $\sigma_{L2}$ system with no inter-subband coupling, the condensate tends to the maximal mixing only as $J_0 \rightarrow \infty$. This is in a stark contrast to the typical metallic-like case, where the mixing is saturated almost immediately after $J_0$ is sufficient to generate any superconductivity. For the details, see the Figures 3 and 4 and the associated text of [25]. In the current model, this $G^0_{\sigma_{L2}}$ distance overcomes the advantage that the $\sigma_{L2}$ had over the $\sigma_{L1}$ coming from the difference in their bandwidths, as described before, leading to the $\sigma_{L1}$ one dominating now the $\sigma_{L2}$ one instead.

### 3.3. The Destructive Interference Parameter Subspace

In this section, the topic of what is the set of the $(J_0, J_1)$ parameters where the destructive interference solution exists prior to the collapse of one of the condensates is discussed. Intuitively, it should be expected that $J_0$ needs to be sufficiently large and $J_1$ sufficiently small, for the $J_0$ to support the generation of the superconducting condensate against the workings of the $J_1$. The BdG equation was solved over the $J_0$ and $J_1$ meshes and the results are presented in Figure 6, for M-M in (a–d) and for M-I in (e–h). The value of

$J_1$, below which the destructive interference solution exists and at which the collapse of the relevant condensate happens, will be further called $J_1^c$. Indeed, $J_1^c$ is a monotonically increasing function of $J_0$, as shown in (a) for $G_{\sigma_{L1}}$ and in (e) for $G_{\sigma_{L2}}^s$. The error bars in the figures correspond to the mesh spacing $\delta J$, as in the obtained results the solution exists for some $n$-th point, but does not exist for the $(n+1)$-st mesh point. To pinpoint the $J_1^c$ exactly is computationally very demanding because of the exponentially increased computational complexity near the critical point, as was explained before. The value $\delta J = 0.125$ meV was adopted as a practical compromise for the 2-dimensional $(J_0, J_1)$ search, as it is sufficiently precise to illustrate the general trend. For example, the trend in (a) is monotonic but clearly not linear. The blue line, which shows the linear fit to the BdG results, describes them very poorly, as opposed to the red line, which shows the parabolic fit. In comparison, the corresponding trend in (e) is much closer to linear, but still in this case the blue line undershoots the $J_1^c$ at the edges of the figure and overshoots it in the central part, clearly missing a non-zero curvature. In the following part of this section, the existence of these curvatures is explored, as well as the difference in magnitude between them.

As was explained before, in the dependence of the subband gap on the $J_1$, for a constant $J_0$, there is the quasi-linear regime for small $J_1$ and the sudden drop power-law regime near the critical point. There are two parameters governing the linear regime: the value of the gap for the uncoupled condensates ($J_1 = 0$) and its derivative at that point. Concerning the former element, the $G_{\sigma_{L1}}(J_1 = 0)$ is shown in Figure 6b and the $G_{\sigma_{L2}}^s(J_1 = 0)$ in (f). Comparison of the two cases shows a visible curvature for the M-M—see the fitted lines in (b)—but the corresponding trend for the M-I is very much linear—see the blue line in (f). In the former case, the gap is located at the Fermi crossing point in the SLBZ, thus for the uncoupled system, it is equal to $J_0 \tilde{\Delta}_{\sigma_{L1}}(J_0)$, where the dependence of the latter factor on $J_0$ was explicitly pointed out. In order for this expression to have a curvature, the second derivative over $J_0$ must not be identically zero. In this case, more specifically $\frac{\partial \tilde{\Delta}_{\sigma_{L1}}}{\partial J_0} + J_0 \frac{\partial^2 \tilde{\Delta}_{\sigma_{L1}}}{\partial J_0^2} > 0$, as the curvature seen in (b) is positive. In the M-I case, there is a non-zero $G_{\sigma_{L2}}^0$ and, as a consequence, $G_{\sigma_{L2}}^s(J_1 = 0)$ takes a more complicated form of $\sqrt{(G_{\sigma_{L2}}^0)^2 + \left(J_0 \tilde{\Delta}_{\sigma_{L2}}(J_0)\right)^2} - G_{\sigma_{L2}}^0$. The exact form of $J_0 \tilde{\Delta}_{\sigma_{L2}}(J_0)$ would depend on the detailed characteristic of the system, but if it is a growing function of $J_0$, then the whole $G_{\sigma_{L2}}^s(J_1 = 0)$ expression grows *slower* as a function of $J_0$. This fact links the flattening of the trend in (f) towards a straight line with the existence of $G_{\sigma_{L2}}^0$. The second parameter characterising the linear regime is the slope of the function, that is $G_{\sigma_{L1}}'(J_1 = 0)$ in the M-M case—see (c)—or $G_{\sigma_{L2}}'(J_1 = 0)$ in the M-I case—see (g), where $' = \frac{\partial}{\partial J_1}$. Qualitatively, the trend is the same in the both cases, specifically of the $\sim \frac{1}{(x-x_0)^c}$ kind, where $0 < c < 1$. This means that the slopes of the quasi-linear parts of the different $J_0$ curves in Figures 2a and 4b are not parallel, but they tend to the horizontal line in the $J_0 \to \infty$ limit. The quantity $J_1^{lc} = \frac{G_{\sigma_{L1}}(J_1=0)}{G_{\sigma_{L1}}'(J_1=0)}$ for the M-M, or $J_1^{lc} = \frac{G_{\sigma_{L2}}^s(J_1=0)}{G_{\sigma_{L2}}'(J_1=0)}$ for the M-I, can be postulated, corresponding to the predicted collapse value $J_1^c$ if the gaps had always followed the initial linear trend. If the initial value of the gaps for the uncoupled system $G(J_1 = 0)$ increased linearly and if the slopes $G'(J_1 = 0)$ were parallel, than the $J_1^{lc}$ would increase linearly with $J_0$. As it really happens, there is a positive curvature in the $J_1^{lc}(J_0)$ trend; not shown for conciseness. The Figures 6d and h show the dependence between the $J_1^{lc}$ and $J_1^c$, for the M-M and the M-I cases, respectively. In both figures, the trend is completely linear, with the parabolic fit not explaining the data visibly better than the blue one, especially in (d). The meaning of this result is that, if the effects coming from the quasi-linear regime are controlled for, there is no additional non-linearity coming from the rapid collapse in Figures 2a and 4b. In fact, the latter effect simply re-scales the $J_1^{lc}$ down to $J_1^c$ by a constant factor. To sum up, the non-linear trend of Figures 2a and 4b can be wholly explained by two factors connected with the quasi-linear regime only, one of which is of purely single-subband origin, as $G(J_1 = 0)$ does not depend on $J_1$, while the another one is of

the inter-subband nature (the derivative over $J_1$). The difference in the $G(J_1 = 0)$ was also identified as a source of the qualitative discrepancy between the M-M and the M-I result.

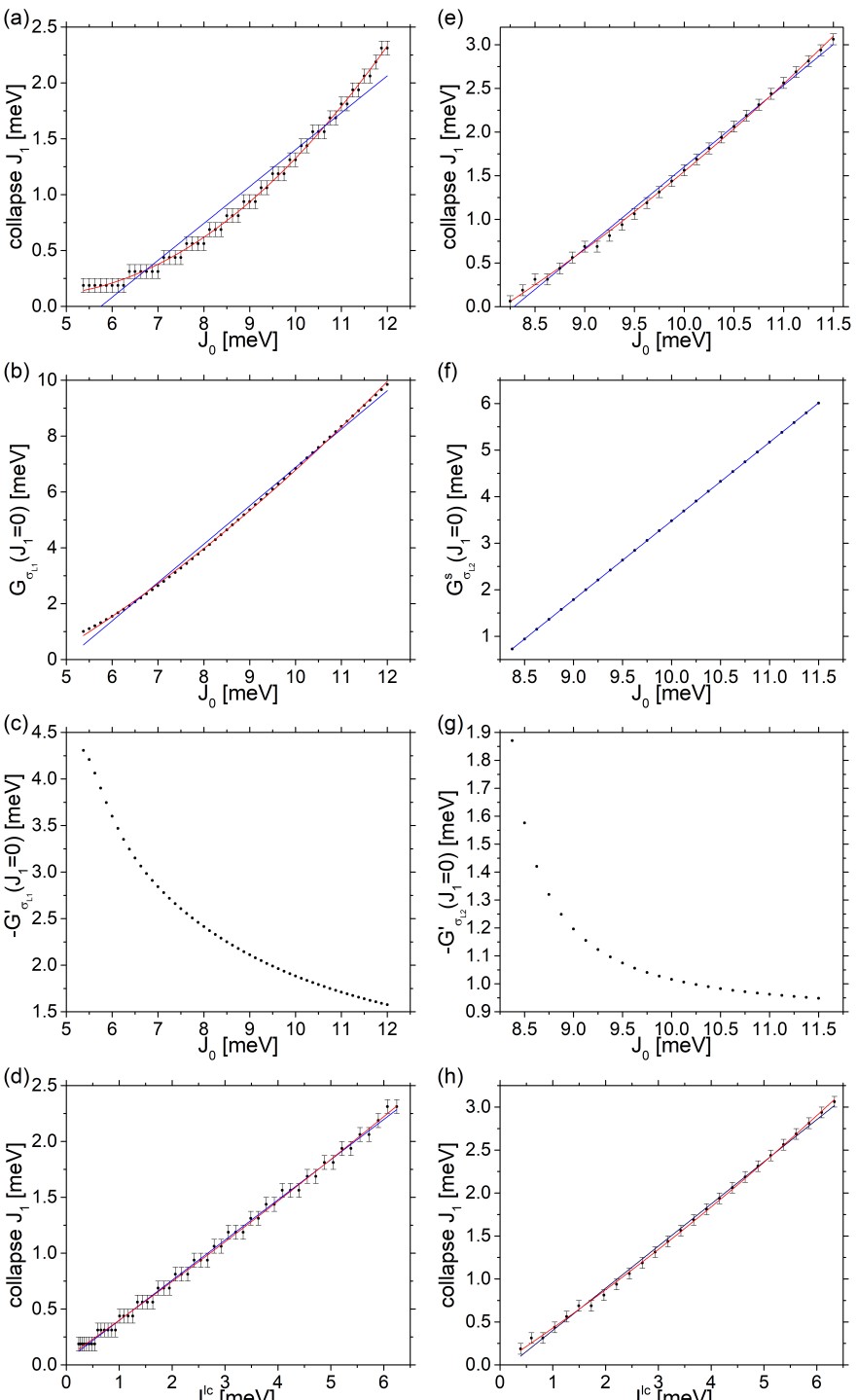

**Figure 6. The M-M.** (**a**) Values of $J_1$ at which $G_{\sigma_{L1}}$ closes and the corresponding condensate collapses, as a function of $J_0$. (**b**) Values of $G_{\sigma_{L1}}$ for the uncoupled system $J_1 = 0$ as a function of $J_0$. (**c**) The derivative of $G_{\sigma_{L1}}$ over $J_1$ for the uncoupled system $J_1 = 0$ as a function of $J_0$. (**d**) Values of $J_1$ at which $G_{\sigma_{L1}}$ closes and the corresponding condensate collapses, as a function of the extrapolated initial linear trend gap closing value $J_1^{lc}$. **The M-I.** (**e**) Analogous to (**a**), but for the $G_{\sigma_{L2}}^s$. (**f**) Analogous to (**b**) but for the $G_{\sigma_{L2}}^s$. (**g**) Analogous to (**c**), but for the $G_{\sigma_{L2}}$. (**h**) Analogous to (**d**), but for where the $G_{\sigma_{L2}}^s$ closes. The points show the results of the BdG simulation, the blue line is the linear fit, and the red line is the parabolic one.

## 4. Discussion

It needs to be especially underlined that—as opposed to many previous works, e.g., [26,28–31]; see also Appendix C for a detailed discussion—this work does not concern whether the intra-subband and the inter-subband couplings are of the same character (attractive/attractive or repulsive/repulsive) or of a opposite character (attractive/repulsive) and, consequently, whether the phases of the partial condensates in the same or the opposite—or $s^{++}/s^{\pm}$ in another notation. In fact, the present work considers the region of the coupling strengths where both the $s^{++}$ as well as $/s^{\pm}$ exists for both the attractive/attractive or attractive/repulsive systems. The authors of one of the previously cited works [31] note that

> It (the relative sign of the gaps on different bands) generally encodes important information about the microscopic pairing interaction, since same-sign gaps (called $s^{++}$ pairing) generally arise from an attractive inter-band interaction, whereas opposite-sign gaps (called $s^{+-}$ pairing) are usually the result of repulsion.

This work studies the frequently omitted opposite case. The *BdG iterative under-relaxation with the phase control* method presented here is the tool for this specific task, easily generalisable to other superconducting systems.

In many publications cited in this work, the enhancement of the superconductivity in multiband systems is attributed to the multiband interaction coinciding with the resonant phenomena of a topological nature. However, the formation of an alternative hypothesis in [32] should be acknowledged.

## 5. Conclusions

In this work, the consequences of the destructive interference of two partial subband condensates were investigated in Q1D SLs in the scope of a simplified model, which includes a single miniband per each subband. It was shown that the destructive solution to the BdG exists in a region where the intra-subband coupling $J_0$ dominates over the inter-subband coupling $J_1$. For a fixed intra-subband coupling $J_0$, two regimes were identified in the dependence of the gap on the inter-subband coupling $J_1$: the dependence is quasi-linear for sufficiently small $J_1$ and then the gap abruptly closes according to the power-law-like function. The internal dynamics of the mentioned regimes were quantitatively described with the help of in-depth analysis of the off-diagonal term in BdG. Two systems were taken into account, one where minibands of both the subbands cross the Fermi level and another one—corresponding to the original fit of the material band dispersions—where one of the minibands has a finite single-particle energy gap. The swap of the dominating partial subband condensate between the two cases is linked to the change from (I) the miniband bandwidth to (II) the separation of the DOS from $E_F$ as the determinant in the system characteristics.

There are several possible ways to develop the model in the future. The most immediate is to include multiple minibands per each subband. A calculation taking into account all the minibands lying in the Debye window was already performed in [25], describing, e.g., the remote inter-miniband resonance, although there was no inter-subband coupling in the latter work. Integrating the miniband–miniband coupling and the subband–subband one into a unified model is the next logical step. The second one is to study the coupling effects between subbands of different types, e.g., the coupling between low-momentum $\sigma$ subbands with the high-momentum $\pi$ subbands in $MgB_2$. In the latter case, the model cannot depend on the single $J_0$ constant to govern over the intra-coupling in two subbands that differ so strongly. Next to mention is the phase difference of the order parameter $\Delta\phi$ in the longitudinal direction, per each primitive SL cell, as described by previous works of the same authors in similar systems [24,25]. It should be noted that any system without any phase difference in the order parameter is necessarily electrically passive, and the enhancement/modulation of the supercurrent flow is one of the more intuitive possible applications of the Q1D SLs.

Finally, the *electron–hole cross-pairing* as opposed to *exchange pair coupling* between different minibands and/or subbands should be addressed. The model as described in this work allows only for a direct pairing (in the sense of the electron- and hole-like mixing to form the BdG solution) at a given $Q$ between the same SL miniband of the same subband. Only the indirect inter-subband coupling is taken into account. In contrast, the implications of the direct *band cross-pairing* were investigated in [27] in the bulk $MgB_2$ and $Ba_{0.6}K_{0.4}Fe_2As_2$. The intraband–crossband regime crossover, gapless state, gap splitting and a temperature dynamic going beyond the BCS were reported. Taking this into account in the model used in the present work seems to be the hardest challenge, as it would require going beyond the Anderson approximation of the BdG, increasing the computational complexity of the model by many orders of magnitude.

An another direction altogether would be the refinement of the description of the underlying crystal structure. The current simple fit, as was mentioned in Section 2, describes the momentum dispersion of each subband as a parabola. The inclusion in this model of a more sophisticated energy–momentum dependence of a single particle is straightforward. Furthermore, the $\vec{k}$-dependent re-scaling of the coupling constants $J_n(\vec{k})$ may be taken into account, especially that the single-particle wavefunctions are already spanned over the set of $(k_x, k_y)$ basis functions.

**Author Contributions:** Conceptualization, W.J.P., M.H.D. and M.Z.M.; methodology, W.J.P. and M.Z.M.; software, W.J.P.; validation, W.J.P.; formal analysis, W.J.P.; investigation, W.J.P. and M.Z.M.; resources, W.J.P. and M.Z.M.; data curation, W.J.P.; writing—original draft preparation, W.J.P. and M.Z.M.; writing—review and editing, W.J.P. and M.Z.M.; visualization, W.J.P.; supervision, M.H.D. and M.Z.M.; project administration, M.Z.M.; funding acquisition, M.Z.M. All authors have read and agreed to the published version of the manuscript.

**Funding:** This research was funded by Office of Naval Research Global grant number N62909-19-1-2130.

**Data Availability Statement:** No data of experimental or statistical nature were used in this work. The computational method is described in detail sufficient to guarantee reproducibility. The custom Fortran 90 code used in the computation will be shared upon a reasonable request.

**Acknowledgments:** We acknowledge the computational facilities provided by LaSCADo (FAPESP grant number 2010/50646-6) and Bruno L.A. Caires (School of Technology/Unicamp) for the institutional support.

**Conflicts of Interest:** The authors declare no conflict of interest. The funders had no role in the design of the study; in the collection, analyses, or interpretation of data; in the writing of the manuscript; or in the decision to publish the results.

## Abbreviations

The following abbreviations are used in this manuscript:

| | |
|---|---|
| MDPI | Multidisciplinary Digital Publishing Institute |
| Q1D | quasi-one-dimensional |
| SL | superlattice |
| BdG | Bogoliubov–de Gennes equations |
| SLBZ | SL Brillouin zone |
| M-M | double-metallic system |
| M-I | metallic-insulating system |
| LHS | left-hand side |
| RHS | right-hand side |
| DOS | density of states |
| BEC | Bose–Einstein condensate |
| BCS | Bardeen–Cooper–Schrieffer [theory/regime] |
| 2D | two dimensional |
| 3D | three dimensional |

## Appendix A. Detailed Description of the BdG Model

In general, the quantities in the Q1D SL model depend on the subband index $S$, on the miniband index $n$, and on the SL wavenumber $Q$, as the symmetry of the envelope wavefunction changes with all three of the mentioned quantum numbers. In the simplified version of the model used in this work, there is only one miniband included per subband, and thus the miniband index is omitted.

*Appendix A.1. The SL Geometry and Material Parametrisation*

This work inherits both the nanostructure and the material model from the previous work of the same group [25]. The material band structure is based on the few-monolayer MgB$_2$; see, e.g., [33]. The nanostructure is a Q1D SL, the longitudinal axis of which runs parallel to the ΓK line of the underlying crystal structure and is called $x$. The system is confined along the perpendicular transverse direction, which runs along the ΓM direction and is called $y$. A single periodic cell of the SL is of the general dimensions $L_X = L_Y = 10$ nm, with the imposed potential energy forming effectively a barrier in the middle-$x$ large-$y$ part of the cell. At the $y$ edges of the cell, the hardwall boundary conditions are used while on the $x$ edges the periodic ones are used.

As mentioned in the main text, several subbands are formed from the $\sigma$ and $\pi$ material bands of MgB$_2$ due to the strong confinement in the width $z$ direction and, furthermore, for each of these material subbands a set of SL miniband ladders is formed by the quantum size effect of the SL itself. The material parameters (band edges and effective masses) were fitted for each material subband and along both the ΓK direction for $x$ as well as the ΓM direction for $y$. The dispersions were fitted by parabolas with vertices at the Γ point, with the help of the data of [33]. For more detailed description of the fit, see Section 3.2 of [25]. The values of the fitted parameters that are relevant to this work are shown in Table A1.

**Table A1.** The values of effective masses and energy offsets used in this work.

| Quantity | Band | Direction: Crystal (SL) | Value |
|:---:|:---:|:---:|:---:|
| $m$ | $\sigma$ lower | M ($y$) | $-0.150$ |
| $m$ | $\sigma$ lower | K ($x$) | $-0.289$ |
| $E_{1,\Gamma}$ | $\sigma$ | | 267.0 meV |
| $\Delta E_\Gamma$ | $\sigma$ | | 95.2 meV |

*Appendix A.2. Single-Particle Model*

The Hamiltonian of the single particle for subband $S$ is

$$H_S = E_{1,\Gamma} + \delta_{S,\sigma_{L2}}\Delta E_\Gamma + \frac{k_x^2}{2m_x} + \frac{k_y^2}{2m_y} + U(x,y). \tag{A1}$$

The smooth potential energy $U$ is effectively equal to zero in the unconstricted region while forming prohibitively large barrier ($U_0 = 1$ keV) in the constricted part. The exact shape of the barrier is defined as product of two one-dimensional factors: $U(x,y;s_X,s_Y) = U_0 f_X(\frac{x}{L_X};s_X)f_Y(\frac{y}{L_Y};s_Y)$, with a single Fermi–Dirac-type function as the $f_Y$ factor and a sum of two of these functions as the $f_X$ factor. The following constraints are imposed:

$$\begin{aligned}
f_Y(0;s_Y) = 0, \qquad f_Y(1;s_Y) = 1, \qquad f_Y(1/2;s_Y) = 1/2, \\
f_X(0;s_X) = f_X(1;s_X) = 0, \quad f_X(1/2;s_X) = 1, \quad f_X(1/3;s_X) = f_X(2/3;s_X) = 1/2.
\end{aligned} \tag{A2}$$

The $s_X$ and $s_Y$ scaling factors have the values such that the $\log_{10}(U[\mathrm{eV}])$ behaves as in Figure 1a.

The imposition of the periodic boundary conditions in $x$ results in the envelope wavefunctions of the

$$\Psi_{Q,S}(x,y) = \Phi_{Q,S}(x,y)\exp(iQx) \tag{A3}$$

kind, with $-\pi/L_X < Q < \pi/L_X$ as the SL wavenumber and the Bloch-like SL function $\Phi_{Q,S}(x,y)$. However, for simplicity $Q$ is given in units of the SLBZ, which is $\frac{2\pi}{L_X}$, so that $\frac{-1}{2} < Q < \frac{1}{2}$. The SLBZ is represented by mesh of $n_Q = 97$ points. The $H_S$ is diagonalised in a set of basis functions

$$f_B\left(\frac{x}{L_X}, \frac{y}{L_Y}; n_X, n_Y\right) = \sqrt{\frac{2}{L_X L_Y}} \exp\left(2i\pi n_X \frac{x}{L_X}\right) \sin\left(\pi n_Y \frac{y}{L_Y}\right), \tag{A4}$$

with $-21 \leqslant n_X \leqslant 21$ and $1 \leqslant n_Y \leqslant 37$, which corresponds to the $k_x$ and $k_y$ lying inside the first *crystal* Brillouin zone.

*Appendix A.3. Anderson Model*

The BdG equations are solved within the Anderson approximation, assuming the electron-like $u(x,y)$ parts and the hole-like $v(x,y)$ parts to be proportional to the single-particle wavefunctions of the corresponding subband $S$ and SL wavenumber $Q$. In this scope, the following 2×2 matrix describes the Hamiltonian part of the BdG model:

$$\begin{pmatrix} \epsilon_{Q,S} & \Delta_{Q,S} \\ \Delta_{Q,S}^* & -\epsilon_{Q,S} \end{pmatrix} \begin{pmatrix} U_{Q,S} \\ V_{Q,S} \end{pmatrix} = E_{Q,S} \begin{pmatrix} U_{Q,S} \\ V_{Q,S} \end{pmatrix}, \tag{A5}$$

where $U_{Q,S}$ and $V_{Q,S}$ are the electron- and hole-like coefficients, $U_{Q,S}^2 + V_{Q,S}^2 = 1$. The mean-field part of the BdG becomes the equation for the matrix elements of the order parameter $\Delta_{Q,S}$:

$$\Delta_{Q,S_1} = J_0 \tilde{\Delta}_{Q,S_1} + J_1 \tilde{\Delta}_{Q,S_1,S_2}; \ S_2 \neq S_1, \tag{A6}$$

$$\tilde{\Delta}_{Q,S_1} = \sum_{Q'} U_{Q',S} V_{Q',S}^* \kappa_{Q,Q',S_1} \left[1 - 2F_D(E_{Q',S}, T)\right] F_D(E_{Q',S} - E_D, T_D), \tag{A7}$$

$$\tilde{\Delta}_{Q,S_1,S_2} = \sum_{Q'} U_{Q',S_2} V_{Q',S_2}^* \kappa'_{Q,S_1,Q',S_2} \left[1 - 2F_D(E_{Q',S}, T)\right] F_D(E_{Q',S} - E_D, T_D), \tag{A8}$$

where $F_D(E, T)$ is the Fermi–Dirac function for energy $E$ and temperature $T$. This work does not investigate the temperature dependence of the system and $T = 0.5$ K is adopted for simplicity. The contribution to the order parameter from the states lying inside the Debye window ($0 < E < E_D = 75$ meV) is switched on, and for the ones outside it is switched off. This is realised with the help of the last term in Equations (A7) and (A8), where a smooth transition with $T_D = 50$ K is assumed. $J_0$ is the constant corresponding to the general strength of the intra-subband coupling while $J_1$ is its inter-subband equivalent. Both have the units of energy.

The Josephson coupling of the $Q, S_1$ and the $Q', S_1$ states of a single subband $S_1$ is described by the dimensionless four-orbital contact term:

$$\kappa_{Q,Q',S_1} = L_X L_Y \iint \Phi_{Q,S_1}^* \Phi_{Q,S_1} \Phi_{Q',S_1} \Phi_{Q',S_1}^* dxdy, \tag{A9}$$

while the equivalent for the inter-subband Josephson coupling between the $Q, S_1$ and the $Q', S_2$ states is

$$\kappa_{Q,S_1,Q',S_2} = L_X L_Y \iint \Phi_{Q,S_1}^* \Phi_{Q,S_1} \Phi_{Q',S_2} \Phi_{Q',S_2}^* dxdy. \tag{A10}$$

In Equations (A5)–(A10), the complex conjugates are included for the sake of generalisation, but in the system under study all the relevant quantities can be taken as real. This is relevant to what follows below, as the phase of the condensate is also a real number.

*Appendix A.4. The Self-Consistent Under-Relaxation with Phase Control*

The Hamiltonian part of the BdG Equation (A5), together with the mean-field part Equations (A6)–(A8), need to be solved self-consistently. In this work, the under-relaxation iterative procedure is used, as described in the following. The required phase control of

the condensates is realised in practice by employing the $\Delta$ phase factor $\chi_S$, as defined in Appendix B. In the case of the constructive interference, it is demanded that $\chi_{L1} > 0$ and $\chi_{L2} > 0$. In the case of the destructive interference, it is demanded instead that $\chi_{L1} > 0$ and $\chi_{L2} < 0$—compare the signs of the corresponding $\Delta$ momentum maps in Figures 3 and 5.

The procedure starts with assuming a random initial condition $U_{Q,S}$ and $V_{Q,S}$ for each possible $Q$ value in the case of each subband $S$, while respecting $U_{Q,S}^2 + V_{Q,S}^2 = 1$. The first phase control is realised at this stage. If any of the mentioned conditions are not met, the phase of the corresponding condensate is swapped with the following substitutions: $V_{Q,S} \to -V_{Q,S}$. Each of the iterations consists of solving Equation (A5) where all of the order matrix elements, both intra- and inter-subband, are the ones given after the previous iteration. This set will be called $\{\Delta\}_{\text{old}}$. This way, the BdG eigenenergies and eigenvectors are obtained in the current iteration. Knowing the latter ones allows in turn for the mean-field equations Equations (A6)–(A8) to be solved, with the result being the new set of order matrix elements, or $\{\Delta\}_{\text{new}}$. The phase control and possible phase swaps are performed at this moment every iteration. This is necessary to exclude the possibility of a random phase wiggling between the $\{\Delta\}_{\text{old}}$ and $\{\Delta\}_{\text{new}}$. Finally, the under-relaxation step is performed by taking the substitution

$$\{\Delta\}_{\text{old}} \to \alpha\{\Delta\}_{\text{new}} + (1 - \alpha)\{\Delta\}_{\text{old}} \tag{A11}$$

for the next iteration. The sufficiently small value of $\alpha$ for the iteration to converge was found by trial and error, being eventually settled at $\alpha = 0.01$.

## Appendix B. Definition of the Momentum Maps of $\Delta$

The starting point for the $\Delta$ momentum maps formula is the definition of the partial subband condensate momentum maps:

$$\bar{\Delta}_{n_X,n_Y,S} = \sum_Q \zeta_{Q,S}(n_X, n_Y)^2 U_{Q,S} V_{Q,S} [1 - 2F_D(E_{Q,S}, T)] F_D(E_{Q,S} - E_D, T_D), \tag{A12}$$

where $\zeta_{Q,S}(n_X, n_Y)$ is the relevant coordinate of the projection of the single-particle wave-function $\Phi_{Q,S}(x, y)$ of Equation (A3) onto the basis function $f_B\left(\frac{x}{L_X}, \frac{y}{L_Y}; n_X, n_Y\right)$ of Equation (A4). The phase factor $\chi_S$ used in the phase control procedure is simply the average of the $\bar{\Delta}_{n_X,n_Y,S}$ over the momentum space:

$$\chi_S = \frac{1}{n_X n_Y} \sum_{n_X, n_Y} \bar{\Delta}_{n_X,n_Y,S}. \tag{A13}$$

Finally, the explicit formula for the $\Delta$ momentum maps is:

$$\hat{\Delta}_{n_X,n_Y,S_1} = J_0 \bar{\Delta}_{n_X,n_Y,S_1} + J_1 \bar{\Delta}_{n_X,n_Y,S_2}; \ S_2 \neq S_1. \tag{A14}$$

This defines a single $\hat{\Delta}_{n_X,n_Y,S}$ momentum space map per subband, allowing the $k_x/k_y$ symmetry to be studied.

## Appendix C. Detailed Discussion about the Character of the Exchange-like Inter-Band Interactions and the Phases of the Partial Condensates

In [26], an SL of quantum wells is employed to model an SL of honeycomb boron layers intercalated by Al and Mg spacers. For a two-band superconductor, the "usual" amplification of the superconducting gaps at the topological Lifshitz transition is predicted when the chemical potential is tuned near the band edge of the second miniband by the quantum confinement effects. However, what is the most relevant from the point of view of the present work, the repulsive character of the exchange-like inter-band interactions is assumed along an attractive intra-band coupling leading to the formation of condensates of opposite signs in the first and second minibands, which is referred to as a $s^\pm$ system.

The direct evidence for the band-interference effects in the form of "deep band" gap minima underlines the importance of the intra-inter coupling ratio control. Nevertheless, Innocenti et al. note that the critical temperature and the absolute values of the gaps does not depend on the sign of the inter-band coupling. This observation should be discussed in the context of the present results. In Equation (6) of [26], there is the $\mathcal{V}_{\mathbf{k},\mathbf{k}'}^{l,l'}\Delta_{l',\mathbf{k}'}$ term, where the $\mathbf{k}$ symbols correspond to the SL momenta in their notation and are irrelevant to the current argument. The $l$ symbols are the band indices. $\mathcal{V}_{\mathbf{k},\mathbf{k}'}^{l,l'}$ correspond to the effective coupling strengths and, as was noted before, in the $l = 1$ version of the aforementioned Equation (6) ibid., the $\mathcal{V}_{\mathbf{k},\mathbf{k}'}^{1,1}$ and $\mathcal{V}_{\mathbf{k},\mathbf{k}'}^{1,2}$ have the opposite signs. Similarly, in the $l = 2$ version, the $\mathcal{V}_{\mathbf{k},\mathbf{k}'}^{2,1}$ and the $\mathcal{V}_{\mathbf{k},\mathbf{k}'}^{2,2}$ have different signs. However, this is compensated by the partial gaps parameters having exclusively the opposite signs, as can be seen in Figures 5 and 6 ibid. with $-\Delta_2/\Delta_1 > 0$. In that case, the $s^{++}$ quantum system—with $\mathcal{V}_{\mathbf{k},\mathbf{k}'}^{1,1}/\mathcal{V}_{\mathbf{k},\mathbf{k}'}^{1,2} > 0$— and $s^{\pm}$ systems—where $\mathcal{V}_{\mathbf{k},\mathbf{k}'}^{1,1}/\mathcal{V}_{\mathbf{k},\mathbf{k}'}^{1,2} < 0$— indeed differ only trivially, as $\mathcal{V}\Delta = (-\mathcal{V})(-\Delta)$. In the terminology used here, they would both be called the constructive interference solution. In contrast, the destructive interference solution would be obtained if $\mathcal{V}_{\mathbf{k},\mathbf{k}'}^{1,1}/\mathcal{V}_{\mathbf{k},\mathbf{k}'}^{1,2} < 0$, $\Delta_2/\Delta_1 > 0$ or, in more direct analogy to the results presented in the present work, if $\mathcal{V}_{\mathbf{k},\mathbf{k}'}^{1,1}/\mathcal{V}_{\mathbf{k},\mathbf{k}'}^{1,2} > 0$, $\Delta_2/\Delta_1 < 0$. The latter solution can be obtained, if it exists, by introducing the phase (sign) control in the iterative solving of the mentioned Equation (6) ibid. and it would be essentially different to the former solution.

A very similar situation takes place in [28] where the possibility of multiband superconductivity in SrTiO$_3$ films and interfaces is investigated using a 2D two-band model, with the chemical potential shifted due to doping or applied electric fields to resonate with the second band. A strong enhancement of the $T_c$ and a sharp feature in the gaps are obtained. Moreover, it is concluded that the intra-band coupling dominates over inter-band coupling. Fernandes et al. make the following observation:

> Our results are independent of the character of the interband interaction: for an attractive (repulsive) interaction, the gaps on two bands have the same (opposite) signs, but the thermodynamic properties discussed here remain the same.

BCS superconductor is studied in [29], in the regime where the Fermi energy is smaller than the Debye energy in the low-density limit. Two opposing non-analytic types of $T_c$ dependence on the low density $n$ are found, depending on dimensionality. A model for multiband systems was also included, relevantly to the present work, in the discussion of which the authors concluded:

> Both attractive and repulsive inter-band interactions increase $T_c$ for two bands (...), as illustrated by the fact that Equation (9) involve only $\bar{\lambda}_{12}^2$: inter-band interactions do not induce inter-band pairing in the present model, but reinforce the intra-band pairing by second-order processes involving the other band.

It is however sufficient to note that there is no intra-band pairing in Equation (1a) ibid., in accordance with the $V_{\beta\beta} = 0$ assumption. In order for any interference to take place, two quantities of variable relative phases are required. In the very well known example of the wave interference, the energy as a measure of interference is proportional to $(A + B)^2$ instead of $A^2 + B^2$. The roles of $A$ and $B$ are played by the $J_0\tilde{\Delta}_{Q,S_1}$ and $J_1\tilde{\Delta}_{Q,S_2}$ terms in our system. A similar effect, to the one found in [29], can also be found in Sections III and IV of [30].

Finally, the experimentally observed suppression of $T_c$ across the Lifshitz transition in doped SrTiO$_3$ with the chemical potential tuned by the carrier concentration was addressed in [31] with the help of a two-band model. In this system, the intra-band attractive coupling is dominating over the repulsive inter-band one. The non-monotonicity was attributed to the strong pair-breaking effect, promoted by disorder, which re-scales the corresponding coupling parameters. The change of the relative signs of the gaps from opposite to the same is reported when moving away from the Lifshitz transition. However, it is not clear whether a transition between the constructive and destructive inter-band interference

happens in that case. It can be noticed that the matrix of the "bare" coupling $\lambda_{ij}$, constants characterising the clean system, needs to be effectively replaced by the following one in the presence of disorder:

$$\left(\tilde{\lambda}_{ij}\right) = \begin{pmatrix} \lambda_{11}\frac{A_{11}^d}{A_{11}^c} + \lambda_{12}\frac{A_{21}^d}{A_{11}^c} & \lambda_{11}\frac{A_{12}^d}{A_{22}^c} + \lambda_{12}\frac{A_{22}^d}{A_{22}^c} \\ \lambda_{21}\frac{A_{11}^d}{A_{11}^c} + \lambda_{22}\frac{A_{21}^d}{A_{11}^c} & \lambda_{21}\frac{A_{12}^d}{A_{22}^c} + \lambda_{22}\frac{A_{22}^d}{A_{22}^c} \end{pmatrix}, \tag{A15}$$

where the $A_{ii}^c$ and $A_{ij}^d$ are elements of the $\hat{A}_{\text{clean}}$ and $\hat{A}_{\text{dirty}}$ matrices, as defined in [31]. While the signs of the $\lambda$ parameters do not depend on the disorder characteristics, the signs of the $\tilde{\lambda}$ ones may change. The comparison between the latter signs and the signs of the gap parameters would yield the answer.

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
