# Peer review of "Destructive Interference of the Superconducting Subband Condensates in the Quasi-1D Multigap Material Nanostructures"

_condensedmatter, doi:10.3390/condmat8010004_

Round 1
Reviewer 1 Report
The paper under review is a work focusing on the effects of constructive and destructive interference of condensates coming from different subbands in a periodic super-lattice.
The results are very interesting and clearly presented.
I strongly suggest to publish the paper as is.
with regards
the reviewer
Author Response
We thank the referee for the review.
Reviewer 2 Report
The submitted manuscript titled "Destructive Interference of the Superconducting Subband Condensates in the Quasi-1D Multigap Material Nanostructures" by W J Pasek M.H. Degani and M Z Maialle describes an interesting case of the interference between condensates in two different spots of the Fermi surfaces in a quasi-one dimensional superconducting superlattice. The authors solve the Bogoliubov-de Gennes equations to get constructive and the destructive interference.
This work is of high interest in the physics of multiple condensates. The paper focus where the destructive and constructive solutions coexist, discussing the importance of destructive solution which is omitted in some works.
The paper is well written, and provides a relevant contribution to a hot topic in the physics of multigap superconductors, therefore it should be published in the present form.
Author Response
We thank the referee for the review.
Reviewer 3 Report
The authors make a computational study of the interaction between subband superconducting condensates in a quasi-one-dimensional nanoribbon system. Such condensates can interact constructively or destructively, and the authors are particularly interested in the latter case, which is studied less often in the literature. The model that they use is inspired by and takes energies from a physical material, few-monolayer MgB2. In the present work, they construct a simplified version of this system, making a toy model with only two subbands that nevertheless captures essential features of a superconducting condensate in a multi-subband system. They use a particular computational method that allows them to obtain the destructive-interference solutions. By examining the dependence of each subband’s superconducting gap on intra- and inter-subband coupling strength, and by making momentum-space maps of the superconducting order parameter, they show to what extent one condensate or the other dominates the overall superconductivity of the system. They find a change in which subband is dominant depending on whether one subband crosses the Fermi level or not.
A strength of this work is that the model system has a useful level of complexity, enough that non-trivial effects occur but not so much that explanations of the effects become poorly constrained. The work is a nice demonstration of the authors’ computational method and lays the groundwork for treating systems that are more closely connected to physical, real-world materials.
The Methods, Results, Conclusion, and Appendices are clear, relevant, and presented in a well-structured manner; the Introduction and Discussion need extensive work to make them concise and clearly connected to the central work of the paper – suggestions are provided in the specific feedback below.
The figures are appropriate and properly show the data. They are generally easy to understand, although the text on Figs. 1(a,c,d), 2, 4, and 6 is very small and does not print well. The font used in Figs. 1(b), 3, and 5 is a good size and prints clearly – I would recommend using it in all of the figures.
Appendix A.1, which describes the setup of the computational model system in detail, makes reference to an earlier work by the authors, Ref. [25]. Since this reference is not open-access and my institution does not subscribe to the journal in which it is published, I do not have access to the complete details of the methods. This may be an issue for other readers as well. If there is critical information in [25], I would recommend briefly repeating it in the Appendix of the present manuscript. Aside from that, I believe that enough detail is given in the Main Text and Appendices to reproduce the results.
Specific feedback:
[Introduction] Lines 88-114, which introduce the main work of the manuscript, are clear, focused, and set up the following sections well. However, the background text that discusses past work from the literature (lines 14-87) is quite long and not clearly oriented towards the central point of the current work. I would suggest condensing each existing paragraph of background into a single summary sentence, probably with a lot of citations, and then making those sentences into a single paragraph of literature review that leads into the paragraphs of lines 88-114.
[Materials and Methods] It was a bit confusing that Fig. 1(b) was mentioned before Fig. 1(a) (lines 199, 128). I would suggest either swapping the panels in the figure or finding some way to introduce Fig. 1(a) first.
[Materials and Methods] For improved clarity, I would suggest re-wording the first mentions of the coupling constants J0 and J1 (lines 126-128) so that the full terminology is explicit: …here we introduce the coupling between them as governed by the inter-subband coupling constant J1, which has the dimension of energy and is a natural counterpart to the intra-subband coupling constant J0.
[Materials and Methods] In the sentence of lines 159-160, the meaning of “relative” was unclear to me. Did you mean relative as in the difference between phonon coupling strengths of subband 1 vs. subband 2, or as in the coupling strength of a single subband in some relative measure?
[Materials and Methods] For improved clarity, on line 191 I would use the term “Q-dispersion” instead of “Q-spectrum” so that the terminology is consistent between the caption of Fig. 2 and the main text.
[Results] I noticed that Fig. 2 used J0 = 10, 11, 12 meV while Fig. 4 used 9, 10, 11 meV. Also, Fig. 3 takes J0 = 11 meV while Fig. 5 takes J J0 = 10 meV. Was there a particular reason for choosing different values between the M-M and M-I cases? I assume that the results would be qualitatively similar if the same J0 were used for both, but it would be good to have explicit reassurance of this and/or a brief explanation of the rationale for choosing those particular values of J0.
[Results] Line 283 mentions a “Debye window.” This term is explained on line 577 in Appendix A.3, but it would help to include a brief explanation in the main text as well.
[Results] Several parts of the paragraph of lines 292-302 are unclear. I found the sentence on lines 294-297 difficult to parse – there were a lot of “only”s to keep track of. It might help to divide it into two separate sentences or to re-order the ideas within it. On lines 299-300, which distance is meant by “this distance”? For clarity, just repeat the full description of the distance that is being referred to here. I would also recommend changing “latter” and “former” (lines 301-2) to explicit descriptions.
[Results] For clarity, I suggest re-writing the sentence of lines 306-8 using full descriptions: Intuitively, it should be expected that J0 needs to be sufficiently large and J1 sufficiently small, for the intra-subband term J0 to support the formation of a condensate against the workings of the inter-subband term J1.
[Results] The discussion of Fig. 6 extends over two paragraphs (first one ending on line 324, next one starting on 325) but “Fig. 6” is only mentioned explicitly in the first. For clarity, I would mention “Fig. 6” again in the second paragraph, e.g. on line 330 in front of the reference to panel (f).
[Results] In the sentence on lines 325-327, I would suggest adding explicit references to the points on Figs. 2(a,b) and 4(a,b) that label the regimes in subband gap vs. J1 that are being mentioned here.
[Results] In the sentence on line 361, instead of saying “The former one” it would be better to explicitly describe what is meant, for added clarity.
[Discussion] The first five paragraphs of the Discussion, lines 364-444, provide a wide-ranging overview of results from the literature that are connected in some way to the current work. I found it very difficult to pick out what the key points were. I wanted to know how the present study improves upon, expands, contradicts, complements, etc. the literature results – this comparison is not clear at the moment. I would suggest summarizing each paragraph into one or two sentences and then making an explicit connection between the summary sentences and the central results of the paper.
[Discussion] The paragraphs discussing future directions for the model, lines 445-470, are clear and helpful. They might be better placed in the Conclusions section – I would suggest moving them to just after line 486, to come after the summary and reminder of the main findings. This would make a nice flow from “what have we done” to “what might we do in the future”.
[Conclusions] The sentence on lines 472-4 refers to “two partial band condensates” but I don’t recognise this terminology from earlier in the paper. For improved clarity, it would be better to use terms that have been used before.
